# SUSD: Structured Unsupervised Skill Discovery through State Factorization

**Seyed Mohammad Hadi Hosseini & Mahdieh Soleymani Baghshah**
Department of Computer Engineering
Sharif University of Technology
{hadi.hosseini17,soleymani}@sharif.edu

## Abstract

Unsupervised Skill Discovery (USD) aims to autonomously learn a diverse set of skills without relying on extrinsic rewards. One of the most common USD approaches is to maximize the Mutual Information (MI) between skill latent variables and states. However, MI-based methods tend to favor simple, static skills due to their invariance properties, limiting the discovery of dynamic, task-relevant behaviors. Distance-Maximizing Skill Discovery (DSD) promotes more dynamic skills by leveraging state-space distances, yet still fall short in encouraging comprehensive skill sets that engage all controllable factors or entities in the environment. In this work, we introduce SUSD, a novel framework that harnesses the compositional structure of environments by factorizing the state space into independent components (e.g., objects or controllable entities). SUSD allocates distinct skill variables to different factors, enabling more fine-grained control on the skill discovery process. A dynamic model also tracks learning across factors, adaptively steering the agent's focus toward underexplored factors. This structured approach not only promotes the discovery of richer and more diverse skills, but also yields a factorized skill representation that enables fine-grained and disentangled control over individual entities which facilitates efficient training of compositional downstream tasks via Hierarchical Reinforcement Learning (HRL). Our experimental results across three environments, with factors ranging from 1 to 10, demonstrate that our method can discover diverse and complex skills without supervision, significantly outperforming existing unsupervised skill discovery methods in factorized and complex environments. Code is publicly available at: https://github.com/hadi-hosseini/SUSD.

## 1 Introduction

Reinforcement Learning (RL) (Sutton et al., 1998) has made significant strides in solving complex tasks, such as strategic games (Schrittwieser et al., 2020; Badia et al., 2020; Mnih et al., 2013) and robotic manipulation (Gu et al., 2016; Andrychowicz et al., 2020), especially when supported by well-shaped, dense reward functions (Booth et al., 2023; Tang et al., 2025; Silver et al., 2018). However, in many real-world settings, rewards are sparse and tasks are long-horizon, making learning much harder.

Therefore, a key limitation is the need to manually design and tune reward functions, which is difficult to scale across multiple tasks (Park et al., 2022). To address this, Unsupervised Skill Discovery (USD) has become widely attended in (Sharma et al., 2019; Eysenbach et al., 2019; Park et al., 2022; 2023; 2024; Campos et al., 2020; Achiam et al., 2018) as an unsupervised RL approach that enables an agent to autonomously explore its environment and acquire a diverse set of useful behaviors. Such pre-trained agents can serve as a general-purpose foundation for efficiently learning a wide range of downstream tasks without relying on task-specific supervision. Precisely, in USD approach, an agent is placed in an environment without any predefined reward function and learns to acquire as many diverse and distinct skills as possible solely through interaction with the environment. Afterwards, these learned skills are leveraged by a high-level policy or zero-shot goal-reaching methods to solve a variety of downstream tasks.

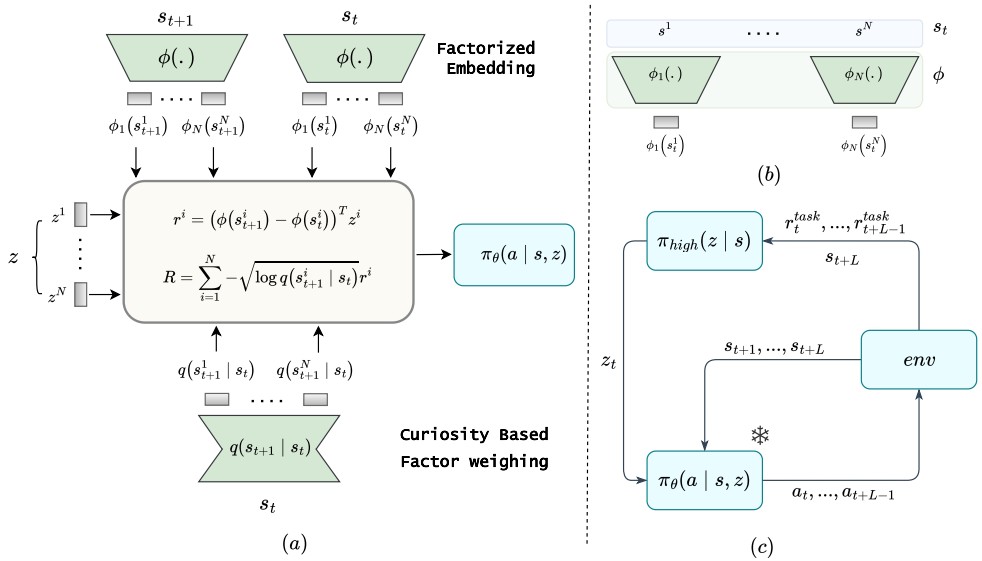

Figure 1: **Illustration of the SUSD Method. (a)** In the skill learning stage, factorized embedding $\phi$ of the current and next state is passed through the corresponding mapping function $\phi(.)$ to obtain a skill latent embedding. Additional details about factorized embedding are shown in **(b)**, where factor $s^i$, $i \in \{1, ..., N\}$, is mapped to its embedding through the function $\phi_i$. These embeddings, together with the skill factor inputs, are used to compute the intrinsic reward $r^i$ of factor $i$. In the curiosity-based factor weighting module, a density model takes the full current state as input and estimates the probability of each next-state factor given the current state $-\log q(s_{t+1}^i | s_t)$. These probabilities are then used as weights to scale the factor-wise intrinsic rewards, which are summed to form the final intrinsic reward for training the skill policy. **(c)** In the task learning stage, the learned skill policy is frozen as a low-level policy, while a high-level policy $\pi_{\text{high}}$ is trained to select a skill $z$ every $L$ steps by maximizing the task reward $r^{\text{task}}$.

Two main branches of USD have gained popularity. The first focuses on MI methods (Eysenbach et al., 2019; Sharma et al., 2019; Kamienny et al., 2022; Kim et al., 2023), that aim to maximize the mutual information between latent skills and states. The second branch centers on DSD methods (Park et al., 2024; 2023; 2022; Kim et al., 2024; Rho et al., 2025), that seek to maximize state changes along skill-specific directions. While MI-based methods often suffer from issues related to invariance to transformations, an aspect discussed in prior work (Park et al., 2024; 2023), DSD methods attempt to encourage more dynamic behaviors by maximizing state-space distances. However, the latent space of DSD methods mostly emphasize only easily controllable factors. Therefore, although state-of-the-art DSD methods perform well in simple environments such as Ant, HalfCheetah, and Walker (Todorov et al., 2012; Brockman et al., 2016; Towers et al., 2024), their performance degrades in more complex environments consisting several controllable elements. This bias arises because DSD lacks explicit mechanisms to ensure skill diversity across all controllable components. As a result, the learned skill latent spaces tend to underrepresent harder-to-control or less immediately responsive elements, limiting the overall expressiveness and coverage of the discovered skills.

To address these limitations, we propose a factorized embedding approach that explicitly leverages the inherent factorization of the environment's state space, an inductive bias commonly present in structured domains such as multi-object environments. Although factorization has been recently explored for skill discovery in MI methods (Hu et al., 2024; Wang et al., 2024), it has not been utilized within DSD methods. In our framework, the state space is decomposed into factors, each corresponding to a distinct subset of controllable features. We associate a dedicated subset of skill variables with each factor, enabling the agent to develop localized skills within each subspace. This decomposed skill representation also naturally supports composability of skills through chaining which makes the learned behaviors more reusable for downstream tasks. However, unlike (Hu et al.,

2024), we do not impose the more restrictive independence assumption. Furthermore, in contrast to existing DSD methods that treat the state space holistically and fail to account for the varying difficulty of controlling different factors, we introduce a curiosity-driven factor weighting mechanism. This component dynamically adjusts the learning emphasis across factors based on the agent's progress, encouraging exploration of underdeveloped or harder-to-control components as others are mastered.

The main contributions of our work are as follows: (1) We propose a novel method for skill learning in environments where the state space can be meaningfully decomposed into subspaces, leveraging this inductive bias to learn a repertoire of skills that collectively cover the entire state space. (2) We design an adaptive weighting mechanism that assigns greater emphasis to less explored factors, encouraging balanced skill acquisition. (3) We evaluate our approach across multiple environments and experiments, demonstrating that our method outperforms leading DSD methods as well as the MI-based simple and factorized approaches.

## 2 RELATED WORKS

### 2.1 UNSUPERVISED SKILL DISCOVERY

USD methods aim to enable an agent to learn a diverse repertoire of skills without relying on pre-defined tasks. These learned skills can later be used to solve downstream tasks either in a zero-shot manner or as components of high-level policy strategies. Research in this area can be broadly categorized into two main directions: (1) Approaches that maximize the MI between skill latent variables and the states. (2) DSD methods, which introduce various distance metrics to explicitly encourage distinguishable behaviors across skills. Our method is built upon the DSD framework.

#### 2.1.1 MUTUAL INFORMATION-BASED SKILL DISCOVER

These methods aim to maximize the MI between states and latent skills, $I(S; Z)$, encouraging different skills to induce distinct and diverse behaviors. This objective is generally intractable, and previous studies approached it using two main strategies: (1) reverse mutual information (Reverse-MI), and (2) forward mutual information (Forward-MI). As regard to Reverse-MI (Mazzaglia et al., 2023; Kamienny et al., 2022; Wang et al., 2024; Kim et al., 2023; Hu et al., 2024; Eysenbach et al., 2019), optimization is in the form of $I(S; Z) = H(Z) - H(Z|S)$, where $H(Z)$ is a constant due to assuming a the skill prior distribution $p(z)$ is fixed. In this approach, a variational lower bound of $I(S; Z)$ is obtained as $I(S; Z) = \mathbb{E}_{z,s}[\log p(z \mid s)] - \mathbb{E}_z[\log p(z)] \geq \mathbb{E}_{z,s}[\log q_\theta(z \mid s)] + \text{const}$ in which a neural network $q_\theta(z|s)$ is typically used to model $p(z|s)$. Regarding Forward-MI (Liu & Abbeel, 2021; Laskin et al., 2022; Sharma et al., 2019), optimization can be expressed as $I(S; Z) = H(S) - H(S|Z)$. To approximate $H(S|Z)$, a neural network is typically used to model a variational distribution. Additionally, maximizing $H(S)$ can be achieved through entropy estimation techniques. A key limitation of these methods is their invariance to scaling or any invertible transformation of input variables. As a result, they tend to learn simple behaviors—such as opening the arm—while failing to capture more dynamic and complex skills (Park et al., 2022; 2023).

#### 2.1.2 DISTANCE-MAXIMIZING SKILL DISCOVERY

Several recent works have adopted DSD to discover diverse and meaningful skills (Park et al., 2022; 2023; 2024; Rho et al., 2025; Kim et al., 2024). DSD utilize the Wasserstein dependency measure as a learning objective for USD, defined as:

$$I_W(S; Z) = \mathcal{W}(p(s, z), p(s)p(z)) \tag{1}$$

where $\mathcal{W}(\cdot, \cdot)$ denotes the Wasserstein distance. In contrast to $I(S; Z) = D_{\text{KL}}(p(s, z)||p(s)p(z))$ in which Kullback–Leibler divergence, $D_{\text{KL}}$, is insensitive to the underlying geometry of the state space, the Wasserstein-based objective explicitly accounts for state-space distances.

DSD methods can intuitively be simplified and expressed as aiming to maximize the state change along the direction specified by the skill $z$ with the following objective (Park et al., 2024):

$$
\sup_{\pi,\phi} \ \mathbb{E}_{p(\tau,z)} \left[ \sum_{t=0}^{T-1} (\phi(s_{t+1}) - \phi(s_t))^\top z \right] \tag{2}
$$
$$
\text{s.t.} \quad \|\phi(s) - \phi(s')\|_L \leq d(s', s), \ \forall (s, s') \in \mathcal{S}_{\text{adj}}
$$

where, $\mathcal{S}_{\text{adj}}$ denotes the set of adjacent state pairs in the Markov Decision Process (MDP), and $L$ is a norm ($L2$ norm is common in prior works).

The main difference between our work and others is that they typically consider relatively simple environments, such as Ant and HalfCheetah (Todorov et al., 2012; Brockman et al., 2016; Towers et al., 2024), where the state can be easily mapped to a low-dimensional latent space via $\phi(\cdot)$. Such simple environments with well-defined mapping from state space to latent space can allow them to easily cover the state space. In contrast, they struggle in more complex, object-centric environments where multiple objects and agents are present and even a simple action can simultaneously affect several factors.

## 2.2 State Space Factorization in RL

Many works leverage knowledge of state factorization for various purposes, including planning (Wang et al., 2022), data augmentation (Pitis et al., 2020), providing intrinsic rewards (Wang et al., 2023; Choi et al., 2024), and goal-conditioned learning (Chuck et al., 2025). Recently, some methods (Hu et al., 2024; Wang et al., 2024; Chuck et al., 2023) have been proposed that consider interactions among objects in the environment and leverage this inductive bias with mutual information objectives to discover diverse skills. For example, DUSDi (Hu et al., 2024) learns disentangled skills where each skill component influences a specific factor, while SkiLD (Wang et al., 2024) induces different interaction graphs to capture object relationships. Although, utilizing the compositional structure of environments help to discover more diverse skills, the existing DSD methods have not yet exploited the state factorization and miss an opportunity to better structure the skill learning process and uncover more comprehensive and composable behaviors.

## 3 Preliminaries and problem setting

An MDP is defined by the tuple $(\mathcal{S}, \mathcal{A}, r, p)$, where $s \in \mathcal{S}$ denotes a state in the state space, $a \in \mathcal{A}$ an action in the action space, $p(\cdot|s, a)$ the transition probability function, and $r : \mathcal{S} \times \mathcal{A} \to \mathbb{R}^+$ the reward function. In this work, we focus on a specialized class of MDPs designed for unsupervised skill discovery in reward-free Factored Markov Decision Processes (FMDPs) (Kearns & Koller, 1999; Osband & Van Roy, 2014; Mohan et al., 2024). There is a long line of prior work (Wang et al., 2024; Hu et al., 2024; Wang et al., 2023; Eysenbach et al., 2019; Pitis et al., 2020; Choi et al., 2024) has adopted Factored Markov Decision Processes (FMDPs) as their environmental framework. In such settings, the state space is structured as a Cartesian product of $N$ subspaces: $\mathcal{S} := \mathcal{S}^1 \times \cdots \times \mathcal{S}^N$, where each $\mathcal{S}^i$ corresponds to a distinct factor of the environment. Each skill is represented by a latent vector $z \in \mathcal{Z}$, where the skill space is factored as $\mathcal{Z} := \mathcal{Z}^1 \times \cdots \times \mathcal{Z}^N$, with each factor $\mathcal{Z}^i \in \mathbb{R}^D_i$. As a result, the overall skill space lies in $\mathbb{R}^{\sum_{i=1}^N D_i}$. A shared skill-conditioned policy $\pi(a|s, z)$ maps states and skill vectors to an action distribution. While the skill space $\mathcal{Z}$ can be either discrete or continuous, we focus on the continuous setting in this work. Nonetheless, our method is readily applicable to discrete skill spaces too as mentioned in Appendix F. To collect a skill trajectory, we sample a skill $z$ from a predefined skill prior distribution $p(z)$ at the beginning of an episode. We then roll out the skill policy $\pi(a|s, z)$ with the sampled $z$ for the entire episode. For the skill prior, we use a standard normal distribution.

In settings where the true state vector is available, the underlying factors can typically be derived directly. When only pixel-based observations are accessible, an encoder can be employed alongside representation learning techniques to extract disentangled factors from the visual input. However, we assume direct access to the underlying state vector that is available in many environments.

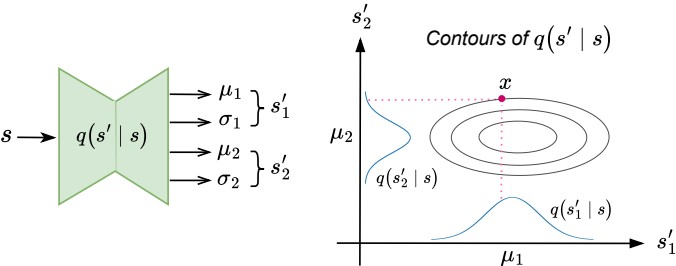

Figure 2: **Left**: The state $s$ is passed to the density model, which estimates the mean and variance of $q(s'|s)$. These statistics are then partitioned by factors to obtain $q(s_i'|s_t)_{i=1}^2$. **Right**: Point $x$ shows high probability in factor 1 but low probability in factor 2—a distinction that cannot be leveraged by the CSD method, which assigns a single weight to the entire state transition rather than to individual state factors.

## 4 METHOD

Existing DSD methods tend to focus on easily controllable state factors and lack mechanisms to encourage diversity across all controllable aspects. To address this, we propose a factorized embedding framework for DSD methods that decomposes the state space into distinct controllable factors, enabling localized and composable skill learning. Additionally, we introduce a curiosity-driven factor weighting mechanism that adaptively emphasizes harder-to-control factors during training.

In this section, we first describe how the observation space, skill latent space, and mapping function are factorized (Section 4.1). Then, we explain how the weight of each factor is computed, which reflects the extent of the agent's struggle with that factor and, consequently, its curiosity to acquire the corresponding skill (Section 4.2). Finally, we present the SUSD method in its entirety, detailing its intrinsic reward, the loss functions used for network updates, and the overall algorithm (Section 4.3).

### 4.1 FACTORIZED DSD

We factorize the observation space into $N$ factors $\{s^i\}_{i=1}^N$, each corresponding to a meaningful component of the environment. In essence, we leverage the compositional structure of the environment as an inductive bias to guide this partitioning of the observation space. More precisely, different controllable elements in the environment (e.g., the target, agent, and ammo in the 2D-Gunner (Lowe et al., 2017)) can be considered as different state factors. Moreover, each skill vector $z$ is composed of $N$ factors, each of dimension $D$. Specifically, for every state factor $s^i$, we associate a latent skill factor $z^i$ within the skill vector. This design enables the agent to learn localized skills specialized to each subspace, improving both disentanglement and control. More precisely, the function $\phi$, which maps the state space to the skill latent space, is structured as $N$ separate networks where each network $i$ takes the $s^i$ as input and outputs $\phi_i(s^i)$. Therefore, the original optimization problem defined in Eq. 2 is converted to the following factorized version:

$$\sup_{\pi, \{\phi_i\}_{i=1}^N} \mathbb{E}_{p(\tau,z)} \sum_{i=1}^N \sum_{t=0}^{T-1} (\phi_i(s_{t+1}^i) - \phi_i(s_t^i))^\top z^i$$

$$\text{subject to} \quad \sum_{i=1}^N \|\phi_i(s'^i) - \phi_i(s^i)\|_2 \leq 1, \quad \forall (s, s') \in \mathcal{S}_{\text{adj}} \tag{3}$$

### 4.2 CURIOSITY-BASED FACTOR WEIGHTING

Inspired by (Park et al., 2023) which prioritizes state transitions that are difficult to achieve under the current skill policy, we aim to encourage the discovery of hard-to-learn behaviors. To this end, we train a density model $q_\theta(s'|s) = \mathcal{N}(\mu_\theta(s), \Sigma_\theta(s))$ on $(s, s')$ tuples collected by the skill policy

and a negative log-likelihood of a transition from the current skill policy, $-\log q_\theta(s_{t+1}|s_t)$, is used as a controllability-aware distance function. It assigns high values for rare transitions while assigns small values for frequently visited transitions.

As discussed in Section 4.1, we factorize the state space into $N$ distinct factors. This raises the question of which factors the agent should attend to at each timestep and how much importance each factor should receive. In this section, we begin with a lemma that allows us to reformulate Eq. 2 so that the distance term is incorporated directly into the objective, rather than appearing in a constraint. Using this lemma, we then derive our final objective function.

**Lemma 4.1.** *In the DSD optimization problem (Eq. 2), we can include the distance term as a coefficient alongside the intrinsic reward. More generally, Eq. 2) can be reformulated as follows:*

$$\sup_{\pi,\phi} \ \mathbb{E}_{p(\tau,z)} \left[ \sum_{t=0}^{T-1} d(s',s)(\phi(s_{t+1}) - \phi(s_t))^\top z \right] \tag{4}$$
$$s.t. \quad \|\phi(s) - \phi(s')\|_L \leq 1, \ \forall(s,s') \in \mathcal{S}_{adj}$$

The proof of this lemma is provided in Appendix A and is adapted from (Kim et al., 2024).

To move beyond the coarse-grained weighting used in (Park et al., 2023), which assigns a single curiosity score per transition, we propose a fine-grained factor-wise weighting mechanism that dynamically adjusts the contribution of each state factor during training. Specifically, to compute the factor-wise curiosity signals $-\log q_\theta(s_{t+1}^i|s_t)$, we feed $s_t$ into the network to obtain $\mu_\theta(s_t)$ and diagonal $\Sigma_\theta(s_t)$ of a multivariate Gaussian distribution. Since the marginals of a Gaussian distribution are themselves Gaussian, we can easily extract the marginal mean and variance for each state factor by partitioning $\mu_\theta(s_t)$ and $\Sigma_\theta(s_t)$ according to the predefined factorization of the observation space. The resulting factor-wise parameters are used to calculate the curiosity weight for each factor as:

$$-\log q_\theta(s_{t+1}^i \mid s_t) \propto (s_{t+1}^i - \mu_\theta^i(s_t))^\top \Sigma_\theta^i(s_t)^{-1} (s_{t+1}^i - \mu_\theta^i(s_t)) \tag{5}$$

The square root of $-\log q_\theta(s_{t+1}^i \mid s_t)$ can be interpreted as a valid distance metric and thus incorporated into the objective defined in Eq. 4 according to Lemma 4.1. The curiosity-based factor weighting module is shown on the bottom of Figure 1. Furhermore, Figure 2 shows how this mechanism provides curiosity-based factor weighting. In this figure, the point $x$ may have a high probability mass under the first factor while it has a low probability mas under the second one. This indicates that more weight should be assigned to the second factor transition than the first one while coarse-grained view of states, as used in CSD (Park et al., 2023), overlook such factor-level controllability.

Accordingly, we our final optimization problem as follows:

$$\sup_{\pi,\{\phi_i\}_{i=1}^N} \ \mathbb{E}_{p(\tau,z)} \sum_{t=0}^{T-1} \sum_{i=1}^N \sqrt{-\log q_\theta(s_{t+1}^i|s_t)} (\phi_i(s_{t+1}^i) - \phi_i(s_t^i))^\top z^i$$
$$\text{subject to} \quad \sum_{i=1}^N \|\phi_i(s'^i) - \phi_i(s^i)\|_2 \leq 1, \quad \forall(s,s') \in \mathcal{S}_{\text{adj}} \tag{6}$$

### 4.3 SUSD TRAINING

We optimize our objective using dual gradient descent (Park et al., 2024; 2023; 2022). That is, with a Lagrange multiplier $\lambda \geq 0$, we use the following dual objectives to train SUSD:

$$r_i^{\text{SUSD}} := (\phi_i(s_{t+1}^i) - \phi_i(s_t^i))^\top z^i, \tag{7}$$

$$J^{\text{SUSD},\phi_i} := \mathbb{E}\left[ r_i^{\text{SUSD}} + \lambda \cdot \min\left(\varepsilon, 1 - \|\phi(s_{t+1}^i) - \phi(s_t^i)\|\right) \right], \tag{8}$$

$$J^{\text{SUSD},\lambda} := -\lambda \cdot \mathbb{E}\left[ \min\left(\varepsilon, 1 - \|\phi(s_{t+1}^i) - \phi(s_t^i)\|\right) \right], \tag{9}$$

$$R := \sum_{i=1}^N \sqrt{-\log q_\theta(s_{t+1}^i|s_t)} r_i^{\text{SUSD}} \tag{10}$$

where $R$ is the intrinsic reward for the skill policy, and $J^{\text{SUSD},\phi_i}$ and $J^{\text{SUSD},\lambda}$ are the objectives for $\phi_i$ and $\lambda$, respectively. We update for each factor independently. The variables $s_{t+1}$ and $s_t$ are sampled from a state pair distribution $p(s_{t+1}, s_t)$ that imposes the constraint in Eq. 6. The slack variable $\varepsilon > 0$ prevents the gradient of $\lambda$ from always being nonnegative. Using these objectives, we train SUSD by optimizing the policy using Eq. 10 as the intrinsic reward, while updating the other components using objectives in Eqs. 8 and 9.

The skill policy $\pi(a \mid s, z)$ is trained with Soft Actor-Critic (SAC) (Haarnoja et al., 2018) according to the obtained reward in Eq. 10 as an intrinsic reward. We train the other components with stochastic gradient descent. We summarize the training procedure of SUSD in Algorithm 1. Implementation details are provided in Appendix E.

---

**Algorithm 1** SUSD: Structured Unsupervised Skill Discovery

---

1: **Initialize** skill policy $\pi(a \mid s, z)$, function $\{\phi_i(s^i)\}_{i=1}^{N}$ conditional density model $q_\theta(s'^i \mid s)$, Lagrange multiplier $\lambda$
2: **for** $i \leftarrow 1$ to #epochs **do**
3:     **for** $j \leftarrow 1$ to #episodes per epoch **do**
4:         Sample skill $z \sim p(z)$
5:         Sample trajectory $\tau$ with $\pi(a \mid s, z)$
6:     **end for**
7:     Fit conditional density model $q_\theta(s'|s)$ using current trajectory samples
8:     Update $\phi_i(s^i)$ with gradient ascent on $J^{\text{SUSD},\phi_i}$ ▷ Eq. 8
9:     Update $\lambda$ with gradient ascent on $J^{\text{SUSD},\lambda}$ ▷ Eq. 9
10:     Update $\pi(a \mid s, z)$ using SAC with intrinsic reward $R$ ▷ Eq. 10
11: **end for**

---

## 5 EXPERIMENTS

In evaluating SUSD, we begin by describing the experimental setup, including details of the environments and baselines (Section 5.1) and then address three key questions: **Q1**: In factorized environments, do our discovered skills outperform other unsupervised reinforcement learning methods on downstream tasks? (Section 5.2) **Q2**: In unfactorized environments, does our method remain competitive with alternative baselines? (Section 5.3) **Q3**: Does our method truly account for all factors during the skill-learning phase? (Section 5.4)

### 5.1 EXPERIMENTAL SETUP

We compare our method against five state-of-the-art unsupervised skill discovery approaches and evaluate these methods in five environments including factorized and unfactorized ones.

**Baselines**: We evaluate our approach against three DSD based methods, namely LSD (Park et al., 2022), CSD (Park et al., 2023), and METRA (Park et al., 2024), as well as DIAYN (Eysenbach et al., 2019) which serve as representative MI-based method and DUSDi (Hu et al., 2024) as a factorized MI-based method. Additional information about these methods have been presented in Appendix D. More details about the hyperparameters are provided in Appendix E.

**Environments**: We evaluate our method on the HalfCheetah and Ant environment (Todorov et al., 2012; Brockman et al., 2016) to demonstrate its applicability to unfactorized settings. The 2D-Gunner is a relatively simple domain, where a point agent can navigate inside a continuous 2D plane, collecting ammo and shooting at targets. Multi-Particle is a multi-agent domain modified based on (Lowe et al., 2017) and this modified version has been introduced in (Hu et al., 2024). In this domain, a centralized controller simultaneously controls 10 heterogenous point-mass agents to interact with 10 stations, where each agent can only interact with a specific station. The Kitchen environment (Zhu et al., 2020; Wang et al., 2023) features a robotic arm and multiple objects, including butter, a meatball, a stove, and a button. In this environment, the agent must learn cookery skills, such as placing the butter or the meatball in the pot. An overview of all environments is shown in Figure 5. The majority of our evaluation focuses on more complex environments, namely

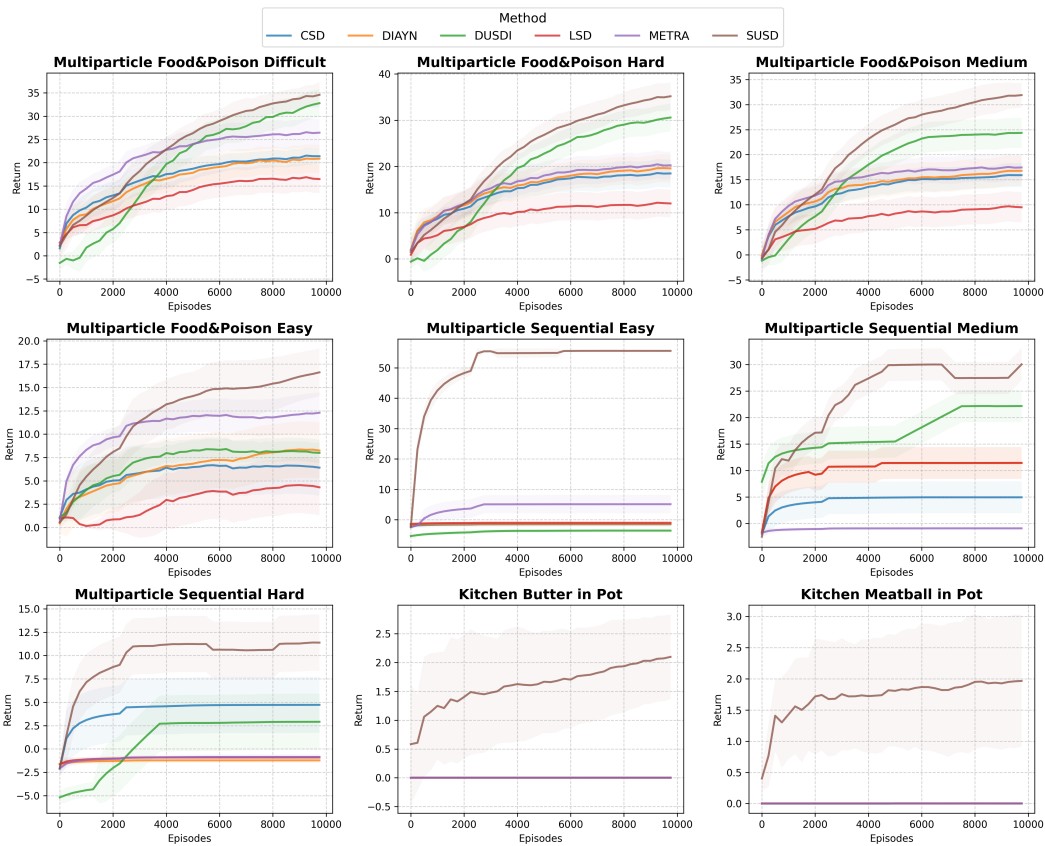

Figure 3: Training curves of SUSD and baseline methods on multiple downstream tasks in the Multi-Particle and Kitchen environments. Each plot shows the mean and standard deviation of returns over 3 random seeds.

Multi-Particle and Kitchen. Further details on the environments and downstream tasks are provided in Appendices B and C, respectively.

## 5.2 EVALUATING FACTORIZED ENVIRONMENTS

In this section, we evaluate the impact of SUSD on factorized environments (**Q1**) by assessing the performance of our method and the baselines across all downstream tasks in the MP and Kitchen environments. We apply a simple factorization strategy where each entity is treated as a distinct factor. Additionally, we can incorporate the agent's state into the entity factors to more accurately capture the causal influence of the agent on the entities in the latent space. Additional details on factorization and further experimental results are provided in Appendix H. As shown in Figure 3, our method generally outperforms all compared methods by a good margin, demonstrating its effectiveness and its ability to leverage the structure of the environment. Moreover, for the Kitchen as a more complex environment (with a high-dimensional observation space), a larger gap is observed. A detailed analysis of the effects of distorted inductive biases in factorization is provided in Appendix M. For an ablation study demonstrating the impact of the curiosity-based weighting mechanism, please refer to Appendix G.

## 5.3 UNFACTORIZED ENVIRONMENTS

To answer (**Q2**), we evaluate the performance of our method in the Ant and HalfCheetah environment (Todorov et al., 2012; Brockman et al., 2016). These environments are relatively simple, consisting of a single agent (e.g. the ant) that can move freely, without any explicit structure. We compare our method to other USD baselines on zero-shot goal-reaching, as described in ME-

TRA (Park et al., 2024) (this technique can be for DSD-based methods). It evaluates the agent's ability to achieve goals without additional training. In this task, the agent is allowed $20K$ steps to accumulate as much reward as possible. To reduce the impact of randomness, we run the experiment across eight different seeds. The results are shown in Figure 6.

## 5.4 RICHNESS OF FACTORIZED LATENT SKILLS

We conduct two experiments to measure the richness of latent skill embedding (**Q3**), In the first experiment, we analyze the Multi-Particle environment and show that SUSD achieves relatively higher state coverage across different factors compared to other baselines. In the second experiment, we demonstrate that our method learns a richer latent skill embedding, capturing a more comprehensive representation of all factors and outperforming other prior DSD-based approaches.

### 5.4.1 STATE COVERAGE ACROSS FACTORS

We evaluate SUSD by randomly selecting a skill every 200 steps and collecting $20K$ rollout steps in the Multi-Particle environment. For each factor (agent), we compute the number of unique states it visits, rounding the agent's positions to determine distinct states. Specifically, following prior works (Park et al., 2022; 2024; 2023), we use the agents' state coverage using their $x$ and $y$ positions. To enable counting, we round each coordinate to two decimal places (e.g., rounding $(0.27392337, -0.46042657)$ to $(0.27, -0.46)$). In Multiple-Particle environment that contains 10 agents, we report the minimum state coverage across agent factors as Worst Agent State Coverage, and the mean state coverage as Average Agent State Coverage. As shown in Figure 4, SUSD achieves substantially better coverage than the baselines, particularly DUSDi. This trend is also observed in 2D-Gunner. This result highlights that SUSD not only learns diverse skills but also ensures balanced exploration across all factors, including the weakest one. We further evaluate binning coverage in Appendix K. Additional comparisons with DUSDi, as another factorized method, are provided in Appendix H.

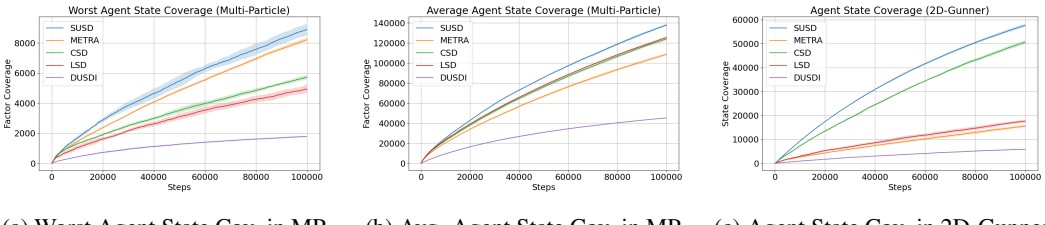

(a) Worst Agent State Cov. in MP    (b) Avg. Agent State Cov. in MP    (c) Agent State Cov. in 2D-Gunner

Figure 4: Comparison of state/factor coverage across different factorized environments.

### 5.4.2 FACTOR DECODING

The objective of this experiment is to demonstrate that our method can learn a latent skill embedding that captures comprehensive, compositional information about all factors. Specifically, mapping observations into the latent skill space via the function $\phi(.)$ should produce embeddings that encode meaningful information for each factor. By training a decoder on top of these embeddings to reconstruct the observations, we can evaluate reconstruction quality using factor-wise MSE. A high-quality latent skill embedding will yield low MSE for each factor, outperforming other baselines and indicating that the embedding effectively captures all relevant information necessary to reconstruct the observations. The factor-wise reconstruction errors from this experiment are reported in Table 1, with details of the training process provided in Appendix J.

Table 1: Factor decoding errors across different factorized environments.

| Method Env | SUSD | METRA | CSD | LSD |
|---|---|---|---|---|
| Multi-Particle | **0.060** | 0.147 | 0.313 | 0.308 |
| Kitchen | **0.014** | 0.028 | 0.049 | 0.038 |
| 2D-Gunner | **0.080** | 0.186 | 0.404 | 0.224 |

## 5.5 EFFECTIVENESS OF LEARNED SKILLS DURING SKILL LEARNING

To evaluate the effectiveness of the learned skills during the skill-learning phase, we designed an experiment that highlights the strength of our method compared to baseline approaches. In the Kitchen environment, we execute 10K rollout steps using the learned skill policy. Every 50 steps, we switch to a randomly selected skill, allowing us to test a wide range of skills. We repeat this experiment 8 times to obtain stable averages. During the rollouts, we measure how often the agent accidentally completes a task (i.e., receives the corresponding task reward) purely by executing these learned skills. The results are summarized in Table 2. For example, on the BiP task, our method achieves an average reward of 39.875 across 8 runs, whereas none of the baseline methods managed to complete this task even once arbitrarily.

Table 2: Average of task rewards obtained through accidental task completions using only the learned skills during the skill-learning phase. Averages are computed over 8 independent runs.

| | SUSD | CSD | METRA | LSD | DUSDi |
|---|---|---|---|---|---|
| BiP (Butter in Pot) | $\mathbf{39.875 \pm 18.452}$ | $0.0 \pm 0.0$ | $0.0 \pm 0.0$ | $0.0 \pm 0.0$ | $0.0 \pm 0.0$ |
| MiP (Meatball in Pot) | $\mathbf{58.875 \pm 25.784}$ | $0.0 \pm 0.0$ | $0.0 \pm 0.0$ | $0.0 \pm 0.0$ | $2.5 \pm 1.14$ |
| PoS (Pot on Stove) | $\mathbf{20.5 \pm 17.965}$ | $0.0 \pm 0.0$ | $0.0 \pm 0.0$ | $0.0 \pm 0.0$ | $1.275 \pm 0.954$ |
| PoT (Pot on Target) | $\mathbf{13.75 \pm 6.923}$ | $0.0 \pm 0.0$ | $0.0 \pm 0.0$ | $0.0 \pm 0.0$ | $0.0 \pm 0.0$ |

## 6 CONCLUSION

Although excellent prior works in unsupervised skill discovery have successfully learned diverse behaviors without supervision, these methods often struggle in complex environments—settings with multiple objects. To address this limitation, we introduced a DSD-based method designed to handle factorized environments by leveraging the environment's structure as an inductive bias to learn diverse skills. We propose a factorized embedding architecture and allocate a subset of skill variables to each controllable factor to avoid underrepresention of harder-to-control elements of the environment in the skill latent space. Moreover, we reformulate the DSD optimization problem and integrate the concept of curiosity-based factor weighting, which dynamically identifies the factors requiring more attention and adjusts the reward weights for each factor accordingly. We empirically demonstrate that SUSD enables agents to acquire diverse and dynamic skills in factorized environments.

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

# A  PROOF OF LEMMA 4.1

We first start with Eq. 2:

$$\sup_{\pi,\phi} \mathbb{E}_{p(\tau,z)} \left[ \sum_{t=0}^{T-1} (\phi(s_{t+1}) - \phi(s_t))^\top z \right] \quad \text{s.t.} \quad \|\phi(s) - \phi(s')\|_2 \le d(s,s'), \ \forall (s,s') \in S_{\text{adj}}. \quad (11)$$

Let the scaled state function be defined as $\tilde{\phi}(s) := \frac{\phi(s)}{d(s,s')}$. Then, we can transform the constraint term in Eq. 11 as follows (since $d(s,s') \ge 0$):

$$\sup_{\pi,\phi} \mathbb{E}_{p(\tau,z)} \left[ \sum_{t=0}^{T-1} (\phi(s_{t+1}) - \phi(s_t))^\top z \right] \quad \text{s.t.} \quad \|\tilde{\phi}(s) - \tilde{\phi}(s')\|_2 \le 1, \ \forall (s,s') \in S_{\text{adj}}. \quad (12)$$

By replacing $\phi(s)$ with $\tilde{\phi}(s) \cdot d(s,s')$ in Eq. 12, we obtain:

$$\sup_{\pi,\phi} \mathbb{E}_{p(\tau,z)} \left[ \sum_{t=0}^{T-1} d(s_t, s_{t+1}) \big(\tilde{\phi}(s_{t+1}) - \tilde{\phi}(s_t)\big)^\top z \right] \quad \text{s.t.} \quad \|\tilde{\phi}(s) - \tilde{\phi}(s')\|_2 \le 1, \ \forall (s,s') \in S_{\text{adj}}.$$
$$(13)$$

# B  ENVIRONMENT DETAILS

## B.1  ANT

As shown in Figure 5(a), the Ant (Todorov et al., 2012; Brockman et al., 2016) environment has an episode length of 200 steps. The observation space consists of a single factor representing the state of the Ant that is 29-dimensional. The action space is continuous, corresponding to the control of the Ant's joints, and has 8 dimensions.

## B.2  HALFCHEETAH

As shown in Figure 5(b), the HalfCheetah (Todorov et al., 2012; Brockman et al., 2016) environment has an episode length of 200 steps. The observation space consists of a single factor representing the state of the cheetah and is 18-dimensional. The action space is continuous, corresponding to the control of the cheetah's joints, and has 6 dimensions.

## B.3  GUNNER

As shown in Figure 5(c), in the 2D-Gunner (Hu et al., 2024) environment, the blue star marks the position of the agent, the blue line marks its shooting direction, the red diamond marks ammo location, and the orange cross marks the target position. The agent has a 18-dimensional observation space, consisting of 3 state factors: Agent Position, Ammo State, Target State. The action is 6-dimensional, 2 for agent movement, 3 for ammo pickup, and 1 for shooting direction.

## B.4  KITCHEN

As shown in Figure 5(d), the Kitchen (Hu et al., 2024; Zhu et al., 2020) environment contains a robot arm, a piece of butter, a meatball, a pot, a stove with its switch, and a target location marked in red. The agent operates in a 4-dimensional action space, while the observation space is 142-dimensional. This observation can be decomposed into seven components: 33 dimensions for the arm, 22 for the pot, 18 for the meatball, 19 for the button, 22 for the stove, and 14 for the target.

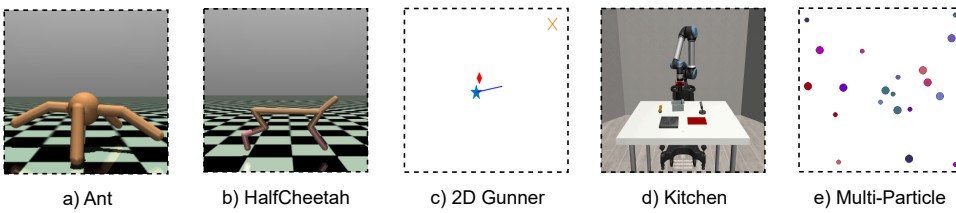

| a) Ant | b) HalfCheetah | c) 2D Gunner | d) Kitchen | e) Multi-Particle |

Figure 5: **Benchmark Environments**

### B.5 MULTI-PARTICLE

As shown in Figure 5(e), agents are represented by small circles, while stations are represented by large circles. Agents can only interact with stations of the same color. The Multi-Particle (Lowe et al., 2017) environment has a 70-dimensional observation space, composed of 10 state factors that capture the states of each agent and its corresponding landmark. The action space is 50-dimensional, with 5 dimensions per agent controlling their movements and interactions with the landmarks.

## C  DOWNSTREAM TASKS

### C.1  ANT

**Multi-goal Ant** (Park et al., 2024): The task requires the agent to reach four target goals, each within a radius of 3. Each goal is randomly selected from the region $[s_x - 7.5, , s_x + 7.5] \times [s_y - 7.5, , s_y + 7.5]$, where $(s_x, s_y)$ denotes the agent's current position in the $x$-$y$ plane. The agent is awarded 2.5 upon reaching a goal. A new goal is generated either when the current goal is reached or if the agent fails to reach it within 50 steps.

### C.2  HALFCHEETAH

**HalfCheetahGoal** (Park et al., 2024): The task is to reach a target goal (within a radius of 3) that is randomly sampled from $[-10, 10]$. The agent receives a reward of 10 upon reaching the goal.

### C.3  GUNNER

**Unlimited Ammo (unlim)** (Hu et al., 2024): In this downstream task, targets appear at random locations, and the agent must approach and shoot each target to score. Since ammunition is unlimited, the agent does not need to collect any.

**Limited Ammo (lim)** (Hu et al., 2024): This downstream task differs from the "Unlimited Ammo" task in that the agent begins without ammunition and must collect it before shooting, while all other aspects remain unchanged.

### C.4  KITCHEN

**Put Butter in the Pot (BiP)**: In this downstream task, the agent's goal is to place the butter in the pot and keep it there. It receives a reward of 1 for each step during which the task is successfully maintained.

**Put Meatball in the Pot (MiP)**: In this downstream task, the agent's goal is to place the meatball in the pot and keep it there. It receives a reward of 1 for each step the task is successfully maintained.

**Put Pot on the Stove (PoS)**: In this downstream task, the agent's goal is to place the pot on the stove and keep it there. It receives a reward of 1 for each step the pot remains on the stove.

**Put Pot on the Target (PoT)**: In this downstream task, the agent's goal is to place the pot on a target position and keep it there. It receives a reward of 1 for each step the pot remains at the target.

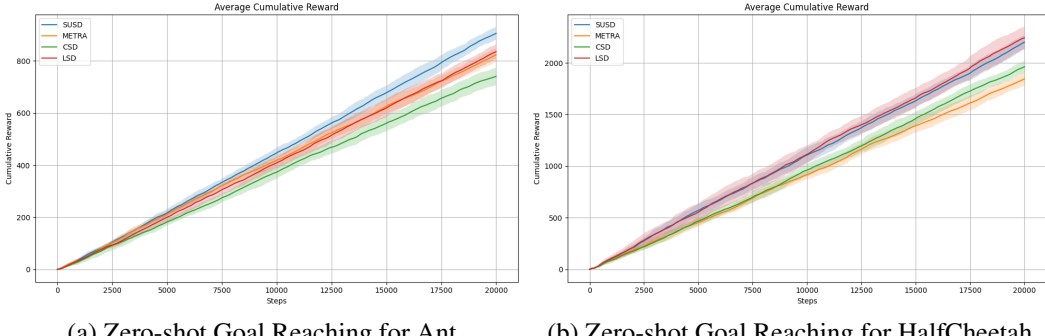

(a) Zero-shot Goal Reaching for Ant      (b) Zero-shot Goal Reaching for HalfCheetah

Figure 6: Zero-shot goal reaching performance of policies learned by skill discovery methods across 8 random seeds. SUSD achieves competitive results in both unfactorized environments.

## C.5 MULTI-PARTICLE (MP)

**Sequential interaction (seq) (easy, medium, hard)** (Hu et al., 2024): In this task, agents are required to interact with their assigned stations following the sequence given by an instruction at the start of each episode. Interacting with stations out of sequence incurs a penalty. In the easy version, the sequence has length 2; in the medium version, length 3; and in the hard version, length 4.

**Food-poison (fp) (easy, medium, hard, difficult)** (Hu et al., 2024): In this downstream task, each station delivers either food or poison to its corresponding agent. Agents must decide whether to interact with their station based on a sequence of binary indicators provided at the beginning of each episode. The easy version uses a sequence of length 2, the medium version length 5, the hard version length 8, and the difficult version length 10.

## D COMPARED METHODS

LSD (Park et al., 2022), CSD (Park et al., 2023), and METRA (Park et al., 2024) are the compared DSD-based methods. LSD measures the distance between states using the $\ell_1$ norm, defined as $d(s', s) = \|s' - s\|$. CSD defines distance in terms of transition probabilities, $d(s', s) = p(s'|s)$. METRA employs temporal distance, given by $d(s', s) = 1$ for the adjacent states in the trajectories. In contrast, DIAYN trains a discriminator to predict the corresponding latent variable $z$ from a given state. Among recent skill learning methods, DUSDi (Hu et al., 2024) leverages the compositional structure of environments by incorporating mutual information (MI) between skill components and state factors into its objective. Specifically, the agent is rewarded for maximizing the MI between each state factor and its corresponding skill component, while minimizing the MI between that component and all other state factors. Although SkiLD (Wang et al., 2024) also adopts environment factorization, it is constrained to environments with a small state space and a limited number of actions. As a result, we were unable to evaluate it on the environments introduced in this work.

## E IMPLEMENTATION DETAILS

**Dimension of the latent space.** For unfactorized environments (i.e., Ant and HalfCheetah), we set the latent skill dimension to $D = 2$ for all evaluated methods. In factorized environments, we use the hyperparameter settings for DUSDi as reported in its paper (Hu et al., 2024). For SUSD, the number of factors is set equal to the number of entities, and we also use skill dimension of $D = 2$ for each factor in all environments. Therefore, we set $N = 3$ in 2D-Gunner (for agent, amno, and target), $N = 7$ in Kitchen (for arm, butter, meatball, button, stove, pot and target entities), and $N = 20$ for Multi-partcle (for 10 agents and 10 stations). When grouping agent and station in this environment, we consider $N = 10$ factors. Appendix I show that higher dimensions of latent space generally do not improve the results of baseline methods (i.e., CSD, METRA) on Multi-Particle environment.

**High-level controllers for downstream tasks.** In Figure 1, we evaluate the learned skills on downstream tasks by training a high-level controller $\pi^h(\mathbf{z} \mid \mathbf{s}, \mathbf{s}_{\text{info}})$, which selects a skill every $K = 5$

Table 3: Hyperparameters for unsupervised skill discovery methods.

| Hyperparameter | Value |
|---|---|
| Learning rate | 0.0001 |
| Optimizer | Adam |
| # episodes per epoch | 8 |
| # gradient steps per epoch | 50 |
| Minibatch size | 256 |
| Discount factor $\gamma$ | 0.99 |
| Replay buffer size | $10^6$ |
| # hidden layers | 2 |
| # hidden units per layer | 1024 |
| Target network smoothing coefficient | 0.995 |
| Entropy coefficient | 0.1 (Adaptive) |
| SUSD $\epsilon$ | $10^{-6}$ |
| SUSD initial $\lambda$ for each factor | 3000 |
| # Number of factors $N$ | 10 (MP), 7 (Kitchen), 3 (Gunner), 1 (Ant, HalfCheetah) |
| # Dimensions of each factor in $\mathbf{Z}$ | 2 |

Table 4: Hyperparameters for downstream policies.

| Hyperparameter | Value |
|---|---|
| # training epochs | $10^4$ |
| # episodes per epoch | 1 |
| # gradient steps per epoch | 50 |
| # skill sample frequency $R$ | 10 |
| Replay buffer size | $10^6$ |
| Target network smoothing coefficient | 0.995 |
| Entropy coefficient | 0.1 (Adaptive) |
| Skill range | $[-1.5, 1.5]^{ND}$ |

environment steps for both MP and Kithchen. At each selection step, the high-level policy chooses a skill $\mathbf{z}$, and the pre-trained low-level skill policy $\pi^l(\mathbf{a} \mid \mathbf{s}, \mathbf{z})$ executes this skill for the next $K$ steps. High-level controllers are trained using SAC (Haarnoja et al., 2018) for continuous skills, with hyperparameters identical to those used in the unsupervised skill discovery methods (Table 4).

**Zero-shot goal-conditioned RL.** In Figure 6, we evaluate the zero-shot performance of our method against other DSD baselines on goal-conditioned downstream tasks in the Ant and HalfCheetah environments. The skill vector $\mathbf{z}$ is recomputed at every step.

We present the full list of hyperparameters used for skill discovery methods in Table 3.

## F    EXTENSION TO DISCRETE SKILL SPACE

For discrete skills, we construct the skill space as $\mathcal{Z} := \mathcal{Z}^1 \times \cdots \times \mathcal{Z}^N$, where $N$ is the number of factors and each $\mathcal{Z}^i$ is a $D$-dimensional one-hot vector, $\mathcal{Z}^i \in \{0,1\}^D$. Although we concatenate this skill vector with the observation to feed into the skill policy, using a one-hot representation for each factor can lead to skill learning collapse. Details on this collapse are discussed in prior DSD-based methods (Park et al., 2022; 2024). To prevent this collapse, we compute the intrinsic reward such that the sum over each factor in the skill vector is zero-mean. This is done using the formula in Eq. 14. Assume that the $k$-th dimension of factor $\mathcal{Z}^i$ is 1:

$$r^i = [\phi(s^i_{t+1}) - \phi(s^i_t)]_k - \frac{1}{N-1} \sum_{j \in \{1,2,\dots,D\} \setminus \{k\}} [\phi(s^i_{t+1}) - \phi(s^i_t)]_j. \tag{14}$$

To evaluate the effectiveness of our discrete approach, we implement the discrete skill space in the 2D-Gunner environment and compare its performance side by side with the continuous skill space,

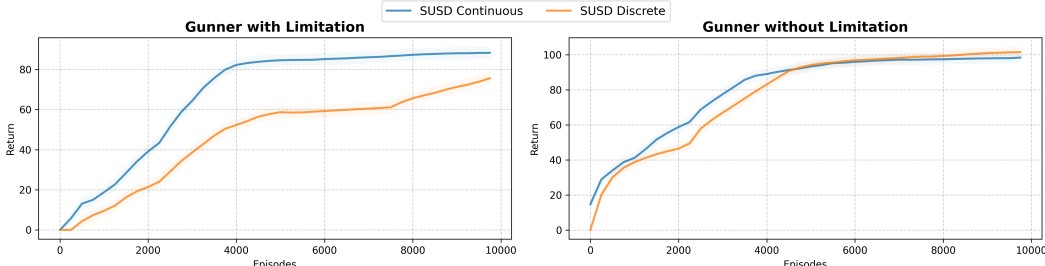

Figure 7: Comparison of discrete and continuous skill spaces in the 2D-Gunner environment.

as shown in Figure 7. As shown, the two methods achieve very similar performance on downstream tasks, indicating that our discrete skill formulation is competitive with the continuous one.

## G ABLATION STUDY

In this study, we evaluate the individual contributions of the curiosity-based weighting and factorization modules by selectively removing them. This allows us to quantify the performance drop when only factorization is applied and when both factorization and curiosity-based weighting are ablated. Specifically, SUSD-w refers to our method without the curiosity-based weighting module, while SUSD-wf refers to our method without both the curiosity-based weighting and factorized embedding. As shown in Figure 8, the performance of these variants decreases compared to our full method which incorporates both modules. This demonstrates that both factorization and curiosity-based weighting are essential for achieving strong results.

## H FACTORIZATION AND FURTHER EXPERIMENTS

DUSDi has carefully engineered factors of the state as mentioned in (Hu et al., 2024). Furthermore, by leveraging knowledge of downstream tasks, the observation space is filtered to focus exclusively on discovering skills that are directly relevant for solving these tasks through HRL. In contrast, we adopt a simpler approach by assigning attributes related to different environmental entities to distinct factors, with the agent factor that can be concatenated to other factors to facilitate the discovery of interactions more easily. In the Multi-Particle environment with 20 objects (agents and stations), we group each agent with its corresponding station as a single factor leads to better performance compared to a factorization treating each object (i.e., agent or station) as a separate factor. Figure 9 compares these two factorizations, showing that incorporating prior knowledge (of requiring each agent to interact with its own station) to align factorization with downstream task structure improves results.

## I IMPACT OF INCREASED SKILL DIMENSIONALITY ON BASELINE PERFORMANCE

Previous USD methods do not exhibit significant performance gains merely by increasing the dimensionality of the skill space. In Figure 10, we show that even when the skill dimensionality of METRA and CSD is increased from 2 to 20, their performance still does not match ours. This highlights a significant gap and demonstrates that leveraging the environment's factorized inductive bias, together with a curiosity-weighted approach, can achieve better performance than merely increasing skill dimensionality. It is worth noting that we use $D = 2$ for all factors across different environments.

## J TRAINING DETAILS FOR FACTOR DECODING

For decoding, we use an MLP having one-hidden layer (with ReLU (Agarap, 2018) activation function) optimized with Adam (Kinga et al., 2015) and mean squared error (MSE) loss for 100

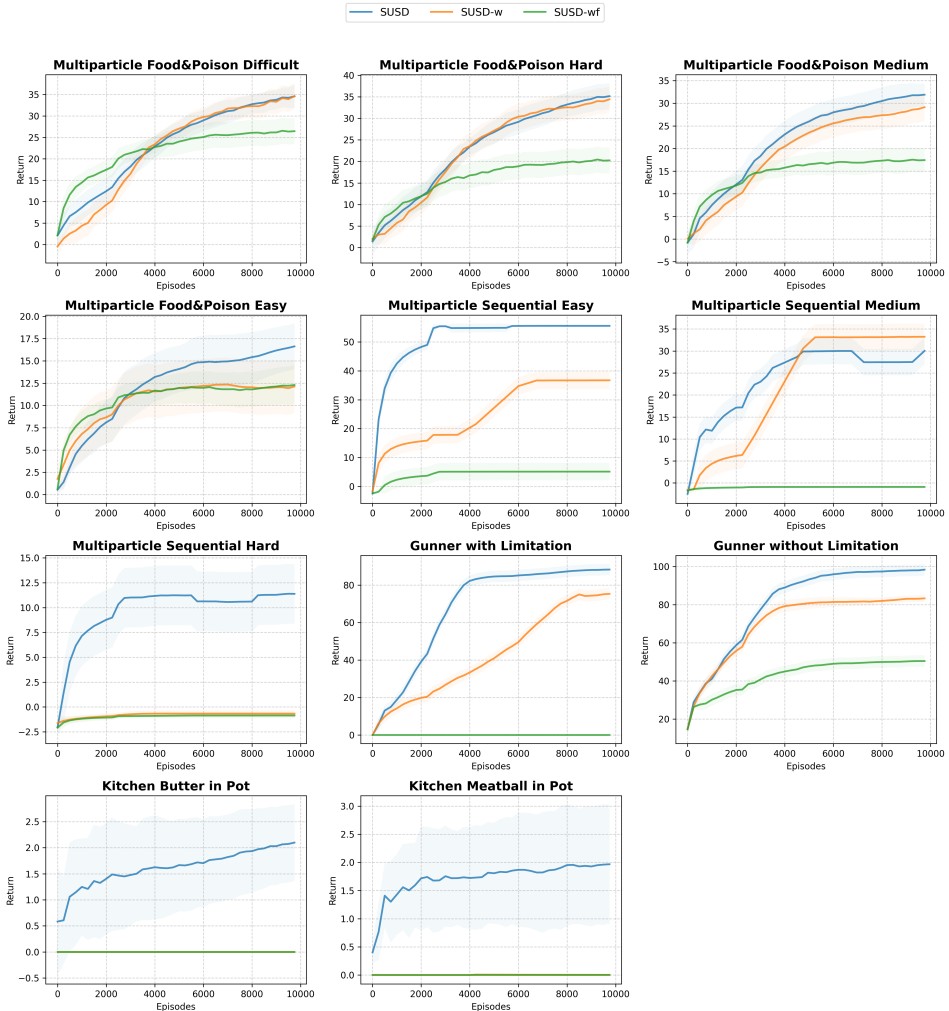

Figure 8: Effect of the curiosity-based weighting module

epochs with learning rate $0.0001$ and batch size $1024$. Training data is collected from $100K$ roll-out steps. Every $200$ steps, we sample a random skill and store the corresponding $(\text{state}, \text{skill})$ pair at each step. We use 80% of the collected data for training and 10% for evaluation and 10% for test. The decoder consists of a hidden layer. We use cross-validation to determine the optimal hidden size for each method. For the Multi-Particle environment, candidate hidden sizes are $\{30, 35, 40, 45, 50, 55, 60, 65\}$. For 2D-Gunner, the candidate values are $\{10, 12, 14, 16\}$ and for the Kitchen environment are $\{20, 30, 40, 50, 60, 70, 80, 90\}$.

## K    BINNING COVERAGE

In this experiment, we aim to evaluate how well our learned skills can cover the state space. To do this, we first divide the $x$ and $y$ axes into $b$ bins each, forming a $b \times b$ grid. We then perform rollouts for a specified number of steps and compute the percentage of grid cells visited by the policy. We conduct this experiment in both the 2D-Gunner and Multi-Particle environments. In the 2D-Gunner environment, we measure bin coverage by discretizing the agent's positions into a $50 \times 50$ grid, resulting in 2500 cells. In the Multi-Particle environment, we use the same setting as outlined above. For each agent, we compute the bin coverage and report both the minimum and average coverage across all agents. The results for this experiment are presented in Figure 11.

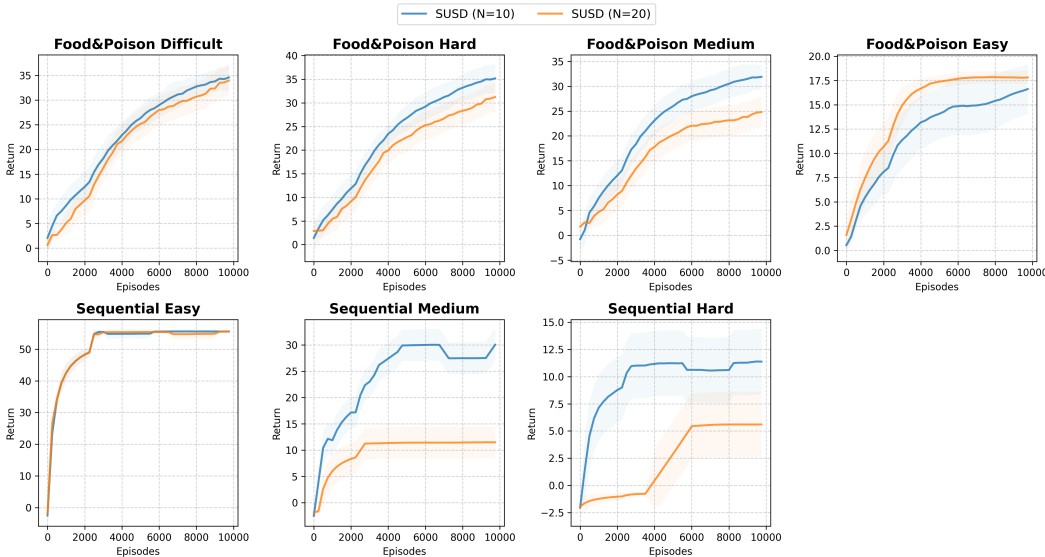

Figure 9: Factorization in Multi-Particle: grouping each agent with its station outperforms treating objects independently.

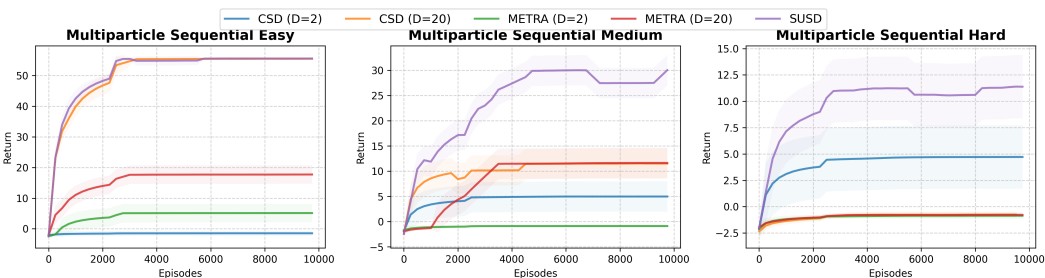

Figure 10: Impact of increasing skill dimensionality on baseline USD methods. Even when METRA and CSD increase their skill dimensionality from 2 to 20, their performance remains below SUSD.

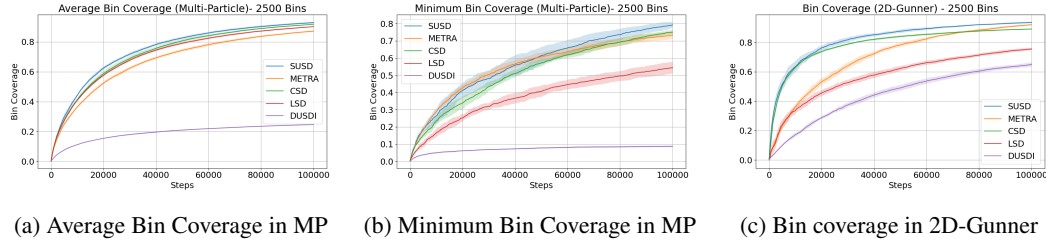

(a) Average Bin Coverage in MP    (b) Minimum Bin Coverage in MP    (c) Bin coverage in 2D-Gunner

Figure 11: Bin coverage comparison. (a) Average bin coverage across factors in MP. (b) Minimum bin coverage across factors in MP. (c) Bin coverage in Gunner environment for 2500 bins.

## L    ADDITIONAL EXPERIMENTS IN THE KITCHEN ENVIRONMENT

We introduce two additional tasks, PoS and PoT, whose definitions are provided in Appendix C. These tasks are added to the Kitchen environment to further evaluate the effectiveness of our method in a 3D setting. The performance of our method compared to the baselines on these tasks is shown in Figure 12.

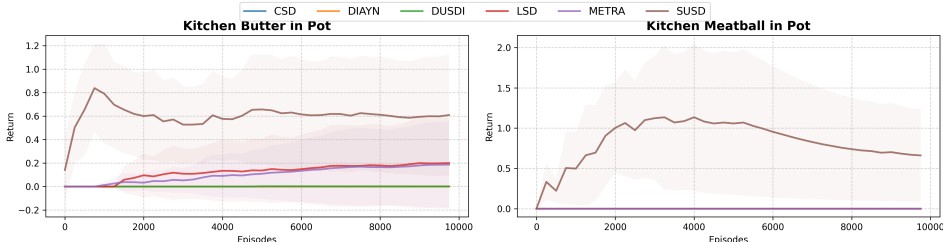

Figure 12: Comparison of SUSD and baseline methods on the PoS and PoT downstream tasks.

# M    ANALYZING OVERFACTORIZATION AND UNDERFACTORIZATION EFFECTS

In environments containing multiple objects, our approach naturally enables the model to encode object-specific information in separate components. Under-factorization reduces the model to a conventional DSD-based formulation that lacks compositional inductive bias and instead learns a single embedding over the entire state vector. Conversely, over-factorization (e.g., decomposing the state into excessively fine-grained components) can introduce misaligned inductive biases and consequently degrade model performance. This occurs because the reward can only incorporate linear combinations of the individual factor embeddings, as in $\sum_{i=1}^{N}(\phi_i(s'^i) - \phi_i(s^i))^T z^i$, whereas embedding dimensions that capture interaction of factors (e.g., $\phi(s^1, s^2)$) may be essential for effectively learning a skill policy. To assess the impact of factorization on performance, we conducted some experiments on the 2D-Gunner environment. In the Gunner environment, the true factors are Agent, Ammo, and Target. Over-factorizing the environment into four factors, by artificially splitting the Agent factor into two random sub-factors, leads to a decline in performance. We also examine under-factorization by reducing the model to two factors, merging the Agent and Ammo factors into a single one. As shown in Figure 13 performance is highest when the true factorization structure is used, and it degrades as more or less factors are imposed.

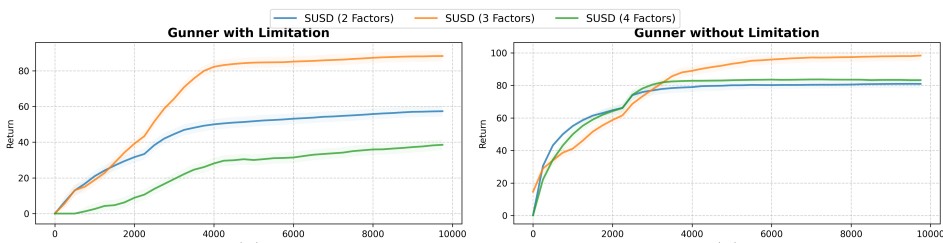

Figure 13: Effect of over/under-factorization on performance in the 2D-Gunner environment.

# N    THE USE OF LARGE LANGUAGE MODELS (LLMS)

We used Large Language Models (LLMs) only in a limited way, specifically for minor writing polish and phrasing suggestions.

