# OpenReview forum: "SUSD: Structured Unsupervised Skill Discovery through State Factorization"
_ICLR.cc/2026/Conference — ICLR 2026 Poster_

### Official Review · Reviewer_1HYa · 2025-10-27

**Soundness:** 2
**Presentation:** 2
**Contribution:** 2
**Rating:** 4
**Confidence:** 2

**Summary:**

This paper has proposed a new unsupervised skill discovery method. Different from the previous mutual information-based and distance-based skill discovery methods, the proposed approach learn skills through state factorization. The proposed approach is evaluated in the Mujoco environment and the MPE environment.

**Strengths:**

1.	This paper is well written, which clearly conveys the importance of the unsupervised skill learning problem.

2.	The proposed approach is extensively evaluated in various domains, such as Mujoco and MPE.

**Weaknesses:**

1.	It is confusing what aspect the state factorization based on, which is related to the definition of skills. As stated in Section 4.2, the skills are learned with the curiosity-based rewards. However, curiosity can vanish with learning, so will the skills collapse to single action? How to prevent the collapse of the skills?

2.	The experiment results demonstrate returns in the downstream tasks. However, as this paper aims at unsupervised skill learning, it is more important to show the comparison of the learned skills among the baselines. Can the skills learned in the Kitchen environment manipulate the task-related objects?

**Questions:**

Please see the weaknesses part.

---

> ### Author Response · Authors · 2025-11-20
>
> We would like to thank you for your helpful comments. Your review raises two concerns, to which we provide a point-to-point response below.
>
> **Response to weakness 1:** In DSD-based skill discovery methods, an embedding space is learned, and the skill policy is conditioned on a skill vector within this space and proposes actions that drive the embedding of the state in the desired direction specified by the skill vector $z$. This is typically framed as a reward function of the form $(\phi(s’)-\phi(s))^Tz$. However, one limitation of this approach is that the embedding space may collapse, often representing only a subset of the factors in the environment—typically those that are more easily controllable.   To mitigate this issue, we exploit the compositional structure of the environment by factorizing the state space into distinct controllable factors or entities and allocating dedicated subspaces of the skill latent space to each. A dynamic model further monitors learning progress across factors, adaptively guiding exploration toward underexplored factors. This structured formulation not only encourages the discovery of richer and more diverse skills but also yields a factorized skill representation that enables fine-grained control over individual entities. As a result, it facilitates more efficient training of compositional downstream tasks within a Hierarchical Reinforcement Learning (HRL) framework.
>
> Curiosity dynamically promotes unexplored factors, and as different factors become sufficiently explored, they receive similar weights. This case mirrors the outcome of ablating factor weighting, where each factor is treated with equal importance.
>
> **Response to weakness 2:** To evaluate the effectiveness of the learned skills during the skill-learning phase, we designed an experiment that highlights the strength of our method compared to baseline approaches.
>
> In the Kitchen environment, we execute 10K rollout steps using the learned skill policy. Every 50 steps, we switch to a randomly selected skill, allowing us to test a wide range of skills. We repeat this experiment 8 times to obtain stable averages. During the rollouts, we measure how often the agent accidentally completes a task (i.e., receives the corresponding task reward) purely by executing these learned skills. The results are summarized in Table 2 of the revised paper, as well as in Table 1 provided below.  For example, on the BiP task, our method achieves an average reward of 39.875 across 8 runs, whereas none of the baseline methods managed to complete this task even once arbitrarily. This experiment is added in Section 5.5 of the paper.
>
> > **Note:** The PoS and PoT tasks are not part of the main paper. We introduced these additional tasks in the Kitchen environment to further assess the effectiveness of our method in a 3D environment, as recommended by Reviewer jqsH.
>
> - **Put Pot on the Stove (PoS):** In this downstream task, the agent’s goal is to place the pot on the stove and keep it there. It receives a reward of 1 for each step during which the task is successfully maintained.
> - **Put Pot on the Target (PoT):** In this downstream task, the agent’s goal is to place the pot on the target location which is defined by red in Figure 5(d) and keep it there. It receives a reward of 1 for each step the task is successfully maintained.
>
> | Task | SUSD | CSD | METRA | LSD | DUSDI |
> |------|------|-----|-------|-----|-------|
> | **BiP (Butter in Pot)** | 39.875 ± 18.452 | 0.0 ± 0.0 | 0.0 ± 0.0 | 0.0 ± 0.0 | 0.0 ± 0.0 |
> | **MiP (Meatball in Pot)** | 58.875 ± 25.784 | 0.0 ± 0.0 | 0.0 ± 0.0 | 0.0 ± 0.0 | 2.5 ± 1.14 |
> | **PoS (Pot on Stove)** | 20.5 ± 17.965 | 0.0 ± 0.0 | 0.0 ± 0.0 | 0.0 ± 0.0 | 1.275 ± 0.954 |
> | **PoT (Pot on Target)** | 13.75 ± 6.923 | 0.0 ± 0.0 | 0.0 ± 0.0 | 0.0 ± 0.0 | 0.0 ± 0.0 |
>
> **Table 1:** Average of task rewards obtained through accidental task completions using only the learned skills during the skill-learning phase. Averages are computed over 8 independent runs.
>
> We thank the reviewer again for the helpful feedback and please let us know if there are any additional concerns or questions.

---

> > ### Comment · Reviewer_1HYa · 2025-11-27
> > **Acknowledgement**
> >
> > Dear Authors
> >
> > Thank you for the thorough rebuttal. I'm relatively satisfied with the updated state of the paper and will increase my score to reflect this.

---

### Official Review · Reviewer_jqsH · 2025-10-27

**Soundness:** 3
**Presentation:** 3
**Contribution:** 2
**Rating:** 6
**Confidence:** 4

**Summary:**

Introduces factorization to distance maximizing skill discovery from mutual-information skill discovery by giving each factor a different skill factor, and then introducing a balancing strategy based on the learning level of the skills for that factor. This is done by working in the FMDP framework and adding a curiosity bonus. Illustrates results on a few factored environments, including 2d modified gridworld and 2d object manipulation.

**Strengths:**

Adds a novel combination of techniques (factored learning to distance-based skill discovery).

Provides adequate empirical evidence to support the method

handles the load-balancing question for factors in a sufficient way.

**Weaknesses:**

Provides marginal change from existing methods in the unsupervised skill discovery space, since both factorization and skill learning have been tried before.

Does not provide clear ablations for which change produces the gain in improvement.

Provides a somewhat limited view of skill learning, focused only on unsupervised skill discovery.

**Questions:**

Skill discovery, even in its modern form, is not introduced by the MISL framework but has been part of hierarchical reinforcement learning for a long time, so the choice of citations is quite narrow for simply calling it "Unsupervised Skill Discovery." The references focus on mutual information based skills.

Another important limitation of using state-space distances is that these distances can often be either hard to learn, such as in the case of temporal distances, or not useful, such as in the case when the state spaces is images. This seems like a more significant issue than simple controllable factors, which is a challenge for both DSD and MISD methods. Thus the motivation does not entirely make sense.

It is probably worth highlighting early in the work that the primary contribution of this work is to apply concepts from existing MISL methods, such as DUSDi or SkiLD, to DSD methods, as factorization and load balancing have been previously addressed.

By assigning each factor a skill parameter, even those that are not controllable, if there is a significant amount of background noise, this could result in poor performance, since the changes from these uncontrollable factors would wash out the reward from the controllable ones.
It seems like curiosity-based factor weighting could result in poor performance in cases where there are factors that change constantly, but are not easy to control. Thus, this relies on a kind of quasistatic assumption where hard-to-control factors are generally stationary, so that getting reward from them will generally be constant until the agent learns to manipulate that factor. This is a challenge with all curiosity-based methods, but moreso in a case where the agent needs to manipulate multiple factors at once.

By giving each factor one skill, this seems like it could fail to load balance factors that require more fine-grained control, and those which require less fine grained control.

This method assumes that all factors are along a flat hierarchy of control: skills learned on one factor are not useful for downstream factors, but this is often not the case (the easy-to-control factors are often useful for downstream factors), and it is not clear if there is a solution for this.

It is not clear why SUSD should outperform any of the methods in the Ant or Halfcheetah environments, yet it appears to do so in Ant. This suggest perhaps a lack of variation or failure to sufficiently optimize baselines.

For factor decoding, doesn't SUSD learn a separate feature map for each factor using ground truth information? Especially since in the FMDP setting the factors are already given. So isn't SUSD given privileged information? A fairer comparison might be a comparison against representations from DUSDi, but that would require significant changes to that algorithm. Thus, it seems unneessary to have this section.

Considering factored MISD methods like SkiLD have been applied to 3D environments like iGibson, it would be good to have a more comprehensive evaluation in a more complex domain to demonstrate that DSD improves performance even in 3D.

---

> ### Author Response · Authors · 2025-11-20
> **Rebuttal - Part 1**
>
> Thank you for your careful reading of our work. We appreciate your comments, as they provide valuable guidance for refining the clarity of our paper. Below, we have addressed each of your points in detail. Please do not hesitate to let us know if any concern remains unaddressed or requires further clarification.
>
> **Response to weakness 1:** While our approach draws inspiration from the factorization ideas in MISL methods, it differs from the disentanglement-based approach of them. Instead of enforcing disentanglement, we decompose the embedding function into components corresponding to distinct factors, without relying on the restrictive assumption of independence. Specifically, we embed each factor separately but don’t enforce the independence of embedded factors which may be restrictive for learning effective policies.  Moreover, to the best of our knowledge, we are the first to introduce dynamic factor weighting in skill discovery, where we explicitly model the importance of each factor throughout the learning process. Notably, methods like DUSDi, even though factorized, do not incorporate factor weighting mechanism.
>
> **Response to weakness 2:** The ablation study was presented in Appendix G. Our method integrates factorization concepts into the DSD approach and also proposes a curiosity-based factor weighting mechanism. In Figure 8, we compared the full method with the version that excludes the dynamic factor weighting mechanism.  Additionally, we had a comparison between our method and METRA (as a DSD baseline that incorporates neither factorization nor curiosity-based factor weighting) in Figure 3. However, we augmented the factor weighting ablation of Figure 8 with the baseline in the revised version to better quantify the contributions of both factorization and dynamic factor weighting to the overall improvement. In fact, a simple DSD method can also be considered as an ablated version of our approach that omits both factorization and consequently factor weighting.
>
> **Response to weakness 3:** All of our related studies (i.e., MISL and DSD methods [1, 2, 3, 4]) have similar viewpoints to ours and focus on discovering diverse behaviors in the absence of extrinsic rewards, which is called unsupervised skill discovery. I have revised the paper to clarify this viewpoint (see Lines 42-48 in the revised version).
>
> **Regarding citation scope:** Thank you for this careful feedback. We revised the second paragraph (lines 42-48) of the introduction to clarify that our focus is on Unsupervised Skill Discovery (USD) and to emphasize the significance of this perspective and the strong engagement of recent studies with it. Consistent with the related works [1, 2, 3, 4], we adopt the terminology “USD” and focus on this problem.
>
> **Regarding limitations of state-space distances:** DSD-based methods such as METRA and CSD do not operate on distances in the raw state space; instead, they rely on distances computed in the learned skill space (i.e., distances of $\phi(s)$ and $\phi(s’)$).
>
> **Regarding highlighting the primary contribution:** Although we have already acknowledged this source of inspiration in Section 2.2, we more prominently emphasized early in the paper that the factorization ideas from existing MISL methods are adapted to DSD approaches (see Lines 100-102 and 106-107 in the revised version of our paper). Furthermore, it is important to note that our method introduces curiosity-based factor weighting during the skill discovery process, a capability not employed in MISL methods like DUSDi.
>
> **Regarding uncontrollable factors and quasistatic assumptions:** Since these factor changes are neither caused by the agent's actions nor by its origin, that is the skill vector $z$, they cannot generally align with the conditioned $z$. As a result, they contribute less to the reward compared to typical curiosity-based intrinsic rewards.   Specifically, $-\log q(s’^i|s)(\phi_i(s’^i)-\phi_i(s^i))^Tz^i$ cannot become large for these factors. More specifically, even if the norm of $(\phi_i(s’^i)-\phi_i(s^i))$ may be large, the difference vector is typically not aligned with the selected skill factor $z_i$, as the state transitions for that factor occur independently of the selected skill vector.   Therefore, the rewards for these factors remain typically low since high rewards require both significant curiosity-based weighting and alignment between $(\phi_i(s’^i)-\phi_i(s^i))$ with $z^i$. Consequently, our method doesn’t suffer from the same limitations as standard curiosity-based intrinsic rewards in promoting exploration.

---

> ### Author Response · Authors · 2025-11-20
> **Rebuttal - Part 2**
>
> **Regarding the limitation of assigning one skill per factor:** Although we assign an equal number of dimensions $D$ to all factors in the skill space for simplicity, the notations in Section 3 indicate that different skills can be represented with varying numbers of dimensions. More importantly, we do not restrict each factor to a single skill; instead, we consider a latent subspace for each factor, within which skills associated with that factor can be discovered.
>
> **Regarding the flat hierarchy assumption:** Popular skill discovery methods, such as DSD and MISL approaches [1, 2, 3, 4, 5], typically assume that factors lie along a flat hierarchy of control. However, from the perspective of Hierarchical Reinforcement Learning (HRL), we can extend this hierarchy to include more than two levels. In this framework, the discovered mid-level skills—which themselves consist of chains of lower-level skills—exhibit the properties you mentioned.  Nevertheless, none of our related work incorporates this multi-level structure. Consequently, consistent with all of the compared methods [1, 2, 3, 4, 5], we pretrain the skill policy first and then use it within a two-level HRL framework to learn policies for downstream tasks.
>
> **Regarding SUSD’s performance in Ant and HalfCheetah:** The methods METRA, CSD, and LSD use Eq. 2 to learn skills, whereas we adopt the reformulated version, Eq. 4 (Lemma 4.1). Although the two forms are mathematically equivalent, our formulation is more straightforward to apply, as we move the distance term from the constraint into the expectation as a weight rather than as a separate constraint.  As shown in Figure 6, our performance in zero-shot goal-reaching tasks in these single-factor environments is competitive but not strictly superior to the other methods. Importantly, for the compared methods, we used the hyperparameters selected by the original methods, and the Ant environment is among those evaluated in these works.
>
> **Regarding factor decoding:** We aimed to show that embedding factors separately in our method helps prevent underrepresenting a subset of factors, where this underrepresentation is more likely when using a monolithic embedding function. However, we can remove this subsection if it seems unnecessary.
>
>
> **Regarding evaluation in complex 3D environments:** Our main paper already includes a challenging 3D environment, the Kitchen environment, which we adopt from ELDEN [6]. We adapted this environment to support low-level actions, as the original version only supports high-level actions such as grasping or turning on.  The Kitchen environment is illustrated in Figure 5(d), and the results of methods on this environment are available in Figure 3. During the rebuttal phase, to further demonstrate the effectiveness of our method in complex 3D settings, we also designed two additional tasks in this environment, named PoS and PoT, which were introduced in Appendix C.4. We also presented the performance of our method relative to the baselines across these two tasks in Appendix L of the revised paper.
>
>
> We thank the reviewer again for the helpful feedback and please let us know if there are any additional concerns or questions.
>
> ---
>
> [1] Seohong Park et al. “METRA: Scalable Unsupervised RL with Metric-Aware Abstraction.” ICLR, 2024.
> [2] Seohong Park et al. “Controllability-Aware Unsupervised Skill Discovery.” ICML, 2023.
> [3] Jiaheng Hu et al. “Disentangled Unsupervised Skill Discovery for Efficient Hierarchical Reinforcement Learning.” NeurIPS, 2024.
> [4] Benjamin Eysenbach et al. “Diversity Is All You Need: Learning Skills Without a Reward Function.” ICML, 2019.
> [5] Seohong Park et al. “Lipschitz-Constrained Unsupervised Skill Discovery.” ICLR, 2022.
> [6] Z. Wang et al. “ELDEN: Exploration via Local Dependencies.” NeurIPS, 2023.

---

### Official Review · Reviewer_ThUn · 2025-10-31

**Soundness:** 3
**Presentation:** 3
**Contribution:** 2
**Rating:** 6
**Confidence:** 4

**Summary:**

This paper introduces SUSD, a new framework for unsupervised skill discovery (USD) that aims to exploit the compositional structure of environments. It builds upon Distance-Maximizing Skill Discovery (DSD) methods like METRA, introducing (1) state space factorization to structure the skill space (similar to DUSDi) and (2) a curiosity-based factor weighting mechanism that prioritizes underexplored factors (similar to CSD). The paper evaluates SUSD across five environments (Ant, HalfCheetah, 2D-Gunner, Kitchen, and Multi-Particle), showing improvements over recent USD baselines, including DIAYN, LSD, CSD, METRA, and DUSDi.

**Strengths:**

The paper proposed a method with strong empirical results, combining the strengths of skill disentanglement (e.g. DUSDi) and Distance-Maximizing Skill Discovery (e.g. METRA). Ablation studies and factorization sensitivity provide useful insight into component contributions.

The factorized embedding formulation is clean and intuitively appealing; it aligns well with factored MDP structure.

The curiosity-based factor weighting is a natural and well-motivated extension to encourage balanced skill learning.

The combination of DUSDi, CSD and METRA is new.

The paper is well-written, where the presentation is clear and easy to understand

**Weaknesses:**

My main concern with this paper is its novelty and conceptual contribution: while the idea of combining state factorization with distance-based skill discovery is sensible, it is not clear how much SUSD goes beyond a straightforward integration of DUSDi, CSD and METRA:
- The factorized skill structure is conceptually almost identical to DUSDi, except that it is applied within a DSD objective rather than a mutual-information one.
- The curiosity-based weighting resembles the controllability weighting from CSD and METRA, but applied per-factor.

Additionally, while the proposed factorization makes intuitive sense in multi-entity environments, the paper doesn’t clearly characterize when this inductive bias is beneficial.
For instance, is the method sensitive to over- or under-factorization? Does performance degrade if the factor decomposition is slightly misaligned with the true controllable entities? A brief sensitivity analysis or discussion would make the claims more general.

**Questions:**

See Weaknesses

---

> ### Author Response · Authors · 2025-11-20
>
> We would like to thank you for your thoughtful review, positive feedback, and constructive comments. Your review raises three important points, which we discuss below.
>
> **Regarding conceptual contribution:** Although our approach draws inspiration from the recent factorization ideas in MISL methods, our factorized model differs from the disentanglement-based approach of DUSDi. Unlike DUSDi, which enforces an independence assumption among factors, our method factorizes the embedding function without relying on the strict and potentially limiting disentanglement assumption. More precisely, rather than learning a monolithic representation $\phi(s)$, we construct the embedding by concatenating the factor-wise components $\phi_i(s^i)$s where $s^i$ is the i-th factor of the state $s$. Specifically, although we embed each factor separately using a compositional inductive bias on the embedding function, we do not enforce independence between these embedded factors as interactions between factors (and so dependencies between their embeddings) may be necessary for learning effective policies. Consequently, we assume only a structured embedding function. Moreover, to the best of our knowledge, we are the first to introduce dynamic factor weighting in the skill discovery process by explicitly modeling the importance of each factor at every stage of skill learning. Notably, even factorized MISL methods such as DUSDi and SkiLD do not incorporate factor weighting.
>
> **Regarding factorization inductive bias:** In environments containing multiple objects, our approach naturally enables the model to encode object-specific information in separate components. Under-factorization reduces the model to a conventional DSD-based formulation that lacks compositional inductive bias and instead learns a single embedding over the entire state vector. Conversely, over-factorization (e.g., decomposing the state into excessively fine-grained components) may introduce misaligned inductive biases and consequently degrade model performance. This occurs because the reward can only incorporate linear combinations of the individual factor embeddings, as in $\sum_{i=1}^N(\phi_i(s’^i)-\phi_i(s^i))^Tz^i$, whereas embedding dimensions that capture interaction of factors (e.g., $\phi(s^1,s^2)$) may be essential for effectively learning a skill policy.
>
>
> **Regarding sensitivity analysis to under/over-factorization:** As discussed above, over-factorization can degrade the method’s performance by introducing a misaligned inductive bias. During the rebuttal phase, we conducted some experiments on the 2D-Gunner environment. In the Gunner environment, the true factors are Agent, Ammo, and Target. Over-factorizing the environment into four factors, by artificially splitting the Agent factor into two sub-factors, leads to a decline in performance. We also examine under-factorization by reducing the model to two factors, merging the Agent and Ammo factors into a single one. As shown in Figure 13, performance is highest when the true factorization structure is used, and it degrades as more or fewer factors are imposed. This experiment is added to Appendix M of the revised paper.
>
>
> We greatly appreciate your time and dedication to providing us with your valuable feedback. If there is anything else that needs clarification or further discussion, please do not hesitate to let us know.

---

> ### Comment · Reviewer_ThUn · 2025-11-26
> **Post-rebuttal**
>
> I appreciate the responses. I think this could be a good contribution to the conference. I will keep my score.

---

> > ### Author Response · Authors · 2025-11-27
> >
> > We thank the reviewer for recognizing our work as a good contribution to the conference.

---

### Official Review · Reviewer_8o7W · 2025-11-01

**Soundness:** 3
**Presentation:** 3
**Contribution:** 2
**Rating:** 4
**Confidence:** 5

**Summary:**

This paper focuses on unsupervised skill discovery and aims to improve distance-maximizing skill discovery performance in environments in multiple controllable objects. To this end, it assumes the state factorization is given and incoroporates the factorization into METRA objective. The proposed method further weights each factorization's contribution to the intrinsic reward based on its prediction accuracy.

**Strengths:**

The paper is well written and easy to follow, especially in the method description and the experiment results.

The proposed method of improving METRA's controllbility over multiple objects, to the best of my knowledge, is novel in the field of unsupervised skill discovery.

The ablations and extra experiment results in the appendix are throughout and provides good insight of each component's importance.

**Weaknesses:**

My major question is the empirical evaluation for the motivation of the paper -- learning skills that engage all controllable factors in the environment.
Specifically, in Sec 5.4.1, since the state space is continuous, the # of unique states is infinite, so the policy may still visit many unique states while only covering a small portion of the state space. It's more appropriate to discretize the state space into bins and see the % of bins that are covered by the policy. I notice the authors mentioned that they "round" the state, kindly correct me if you are already doing that (would be helpful to explain how exactly the rounding is conducted). In addition, visualizing the each factor's coverage is another valid way to show better controllability.

Minor:
missing space on Line 478.

**Questions:**

Different state factors can have different scales and thus different $-\log(s^i_{t+1}|s^i_t)$ scales, how do you account for this scale differences when weighting the intrinisc reward?

---

> ### Author Response · Authors · 2025-11-20
>
> Thank you for your helpful feedback. We greatly appreciate your comments, which will help us further improve the clarity and quality of the paper. We address your points below, and please do not hesitate to let us know if any concern remains unaddressed or requires further clarification.
>
> **Response to weakness:** It is infeasible to perform this experiment directly in the full state space due to its continuous and high-dimensional nature. Therefore, following prior works [1, 2, 3], we use the agents’ x,y positions to approximate their state coverage. To make counting possible, we round the coordinates to two decimal places which was clarified in the revised version of our paper (see Lines 437-439). Moreover, in Figure 4(b), we count the number of unique (x,y) positions per agent (across a total of 10 agents) and report the average number of unique positions. Figure 4(a) presents a similar analysis but shows the minimum number of unique positions among all agents, highlighting the coverage of the least-explorative agent. Finally, Figure 4(c) applies the same procedure to a different environment (2D-Gunner) and reports the number of unique positions visited by the gunner.
>
> Furthermore, in the rebuttal phase, we tested the binning approach you proposed. To do this, we first divide the $x$ and $y$ axes into $b$ bins each, forming a $b \times b$ grid. We then perform rollouts for a specified number of steps and compute the percentage of grid cells visited by the policy. We conduct this experiment in both the 2D-Gunner and Multi-Particle environments and report its results in Appendix K. More precisely, we showed the computed bin coverage for 2D-Gunner and both the minimum and average coverage across all agents for the Multi-Particle environment in Figure 11.
>
> **Response to minor weakness:** Thank you for catching the missing space. We fixed it in the revised version.
>
> **Response to question:** In Eq. (5), the distances are expressed as Mahalanobis distances. Under the assumption of a diagonal covariance matrix as discussed in our work, this implies that the squared distance along each dimension is normalized by the corresponding variance. Consequently, the scale differences among state dimensions do not affect the weighting term.
>
>
> We thank the reviewer again for the helpful feedback and please let us know if there are any additional concerns or questions.
>
> ---
>
> [1] Seohong Park et al. “METRA: Scalable Unsupervised RL with Metric-Aware Abstraction.” ICLR, 2024.
> [2] Seohong Park et al. “Controllability-Aware Unsupervised Skill Discovery.” ICML, 2023.
> [3] Seohong Park et al. “Lipschitz-Constrained Unsupervised Skill Discovery.” ICLR, 2022.

---

> > ### Comment · Reviewer_8o7W · 2025-11-26
> >
> > I appreciate the extra results from the authors. I view the paper positively and raised my score. Yet, in Appendix K, the improvement of SUSD over baselines seem marginal, and I encourage the authors to add corresponding discussions.

---

> > > ### Author Response · Authors · 2025-11-27
> > >
> > > We thank the reviewer for the positive assessment and for raising the score. We appreciate the suggestion and will add a clear discussion for Appendix K in the revised paper.

---

### Author Response · Authors · 2025-12-01
**Global Response**

**Dear AC,**

Thank you for taking over the evaluation of our submission following the recent policy changes. We aim to provide a concise and transparent summary of the rebuttal and discussion phase, providing a clear overview of the original review process and responses.

---

**Summary of Rebuttal and Discussion**

During the original rebuttal period—before reviewer identities were leaked and scores were reverted—we engaged thoroughly with all reviewers’ comments. Our responses are summarized below:

**Reviewer 8o7W**

- **Experiments:**

   - **Bin state coverage experiments:** Bin-based coverage results in the 2D-Gunner and MP environments were added to Appendix K.

- **Clarifications:**

  - **State rounding:** We explained how continuous states are rounded for coverage evaluation.

   - **Factor scaling:** We discussed how scale differences across factors are handled in the proposed curiosity-based weighting formulation.

**Reviewer ThUn**

- **Experiments:**

    - **Sensitivity analysis of over-/under-factorization:** Experiments with different numbers of factors in 2D-Gunner to study over-/under-factorization were added to Appendix M.

- **Clarifications:**

   - **Conceptual contribution:** The distinction between our factorization-based method and DUSDi that is based on disentanglement (which is a stronger and more restrictive assumption) was clarified. Moreover, we highlight that our dynamic factor-weighting mechanism is a novel contribution that has not been explored in prior work.

    - **Factorization inductive bias:** We elaborated on our use of a straightforward factorization method, in which each entity is treated as an individual factor, and examined the effects of both over- and under-factorization on the outcomes.

**Reviewer jqsH**

- **Experiments:**

    - **Additional ablation studies:** The full method with (i) versions excluding the dynamic factor weighting mechanism and (ii) the METRA baseline (without both factorization and factor weighting) was compared in Appendix G.

   - **Additional downstream tasks:** To demonstrate performance in complex 3D environments, downstream tasks in the 3D Kitchen environment were expanded and the results were added to Appendix L.

- **Clarifications:**

   - **Contribution and related work:** We clarified SUSD’s novelty relative to MISL methods which are discussed above in response to **Reviewer ThUn** too.

   - **Uncontrollable factors and quasistatic assumptions:** We explained how limitations of standard curiosity-based intrinsic rewards are alleviated here since the factors that are not controlled by the agent minimally affect the reward.

   - **Skill hierarchy:** We discussed the potential extension to a multi-level HRL framework for skill composition. However, it is important to note that none of the related USD work has explored this case.

   -  We also clarified certain aspects of our method that were misunderstood by the reviewer, including:

         - **State-space distances:** Our approach relies on latent space distances rather than state-space distances mentioned by the reviewer.

         - **Assigning one skill to each factor:** We do not assign a single skill to each factor. Instead, we define a latent subspace for each factor, within which skills associated with that factor can be discovered.

**Reviewer 1HYa**

 - **Experiments:**

   - **Skill-learning phase evaluation in Kitchen:** Accidental completion of tasks (BiP, MiP, PoS, PoT) by randomly sampling skills, assessing both effectiveness and coverage, was measured and reported in Section 5.5 of the main paper.

- **Clarifications:**

    - **Preventing skill collapse:** We explained how factorized state spaces and dynamic factor weighting ensure diverse skills and prevent collapse as curiosity diminishes.

---

**Reviewers’ Post-Rebuttal Assessment (Before Score Reset)**

Before the new AC was assigned and scores were reset due to the policy change:

Reviewer **8o7W** increased their score from 4 → 6, noting that they “view the paper positively” after our rebuttal and added experiments.

Reviewer **ThUn** maintained their score at 6 and stated that they “think this could be a good contribution to the conference.”

Reviewer **jqsH** did not participate in the rebuttal discussion before the policy change.

Reviewer **1HYa** increased their score from 4 → 6, acknowledging that they were “satisfied with the updated state of the paper.”

---

**Closing Remarks**

We hope that the additional experiments, clarifications, and analyses submitted in the rebuttal phase convey the strength and contribution of our work.

Thank you very much for your time and for evaluating our submission.

---

### Meta-Review · Area_Chair_b2wb · 2026-01-06

**Summary:**

This paper introduces a novel skill discovery method based on factored learning.

Strengths

* Novel combination of factorized learning with skill discovery.

* Strong empirical results

Concerns

* See `Reviewer Concerns`

The authors have addressed the reviewers’ concerns in the rebuttal, and I believe this paper makes a meaningful contribution that will be valuable to the community.

**Reviewer Concerns:**

### Reviewer 8o7W

* [resolved] Empirical evaluation for the motivation: added bin-based coverage results in the 2D-Gunner and MP environments

### Reviewer ThUn

* [resolved] Novelty and conceptual contribution

### Reviewer jqsH

* [resolved] Novelty

* [resolved] Ablation study

### Reviewer 1HYa

* [resolved]  Definition of skills

**Reviewer Scores:**

* Reviewer 8o7W: 4 $\rightarrow$ 6

* Reviewer ThUn: 6 $\rightarrow$ 6

* Reviewer jqsH: 6 $\rightarrow$ 6

* Reviewer 1HYa: 4 $\rightarrow$ 6

---

### Decision · Program_Chairs · 2026-01-26

Accept (Poster)